# Preparation of a Novel CO$_2$-Responsive Polymer/Multiwall Carbon Nanotube Composite

**Yonggang Ma [1],\*, Xin Chen [1], Dehui Han [1], Zhe Zhao [2] and Wenting Lu [1]**

[1] College of Pharmacy, Taizhou Polytechnic College, Taizhou 225399, China; chenxinchauncey@163.com (X.C.); dehuihan@gmail.com (D.H.); luwenting0702@163.com (W.L.)
[2] Center for Electron Microscopy and Tianjian Key Lab for Advanced Functional Porous Materials, Institute for New Energy Materials & Low-Carbon Technologies, School of Material Science and Engineering, Tianjin University of Technology, Tianjin 300384, China; zhaozhe7633@163.com
\* Correspondence: ma760727@163.com

**Abstract:** A CO$_2$-responsive composite of multiwall carbon nanotube (MWCNT) coated with poly-dopamine (PDA) and polydimethylamino-ethyl methacrylate (PDMAEMA) was prepared. The PDA was first self-polymerized on the surface of carbon nanotube. 2-bromoisobutyryl bromide (BiBB) was then immobilized by PDA and then initiated the ATRP of DMAEMA on the carbon nanotube surface. The resulting composite was characterized by Fourier-transform infrared spectroscopy (FTIR) and transmission electron microscopy (TEM). The CO$_2$-responsive test was performed by bubbling CO$_2$ into the mixture of MWCNT-PDA-PDMAEMA composite in water. A well-dispersed solution was obtained and the UV-Vis transmittance decreased dramatically. This is attributed to the reaction between PDMAEMA and CO$_2$. The formation of ammonium bicarbonates on the surface of carbon nanotubes leads to the separation of nanotube bundles. This process can be reversed as the removal of CO$_2$ by bubbling N$_2$.

**Keywords:** CO$_2$-responsive; polymer; multiwall carbon nanotube (MWCNT); composite; ATRP

## 1. Introduction

Switchable systems are of great interest for green chemistry research and triggers such as temperature, light, voltage, acids, bases, oxidants or reductants are limited by their cost and environmental concern [1]. CO$_2$, however, is abundant, inexpensive, stable and non-toxic to the reaction materials, which makes it an appropriate trigger for switchable systems [2]. The first switchable-polarity solvent (SPS) was prepared by Jessop and coworkers, using amidine/alcohol or guanidine/alcohol mixtures [3]. The polarity of the solvent is switched from low polarity to high polarity upon CO$_2$ gas passing through and the polarity is reversed by removing CO$_2$. The proposed mechanism is the reaction of CO$_2$ with alcohol and amidine, which results in the formation of ionic liquids. After this, a variety of different SPS systems have been developed for the application of reaction media [4,5], extraction media [6,7], CO$_2$ detection [8], and CO$_2$ capture [9–14]. A series of switchable hydrophilicity solvents (SHS) have been prepared to avoid the distillation of volatile solvents [15,16]. In addition, scientists have also developed switchable surfactants, solutes, as well as catalysts for different applications [17–19].

Another category of switchable system is CO$_2$-responsive polymers. One of the earliest applications of CO$_2$ was to recover solid from a pH responsive latex in 1986 [20]. Most of the polymers have switchable functional groups such as tertiary amine, amidine, guanidine, imidazole, or carboxylic acid, which are capable of changing from neutral to cationic, anionic, or carbamate salts [2]. However, the synthesis condition is strict and the amidine-containing polymers are inherently hydrolytically unstable [21]. Instead of functionalizing the polymer itself, Han et al. proposed a more general strategy to incorporate an amine-containing monomer such as N,N-dimethylaminoethyl methacrylate (DMAEMA) into non-

$CO_2$-responsive, thermosensitive polymers [22]. The resulting copolymer consequently shows a $CO_2$-switchable lower critical solution temperature (LCST). The switchable LCST makes it possible to design different smart materials with reversible structure triggered by $CO_2$ [23–26].

Since the synthesis of carbon nanotubes (CNTs) [27], scientists have found a wide range of applications for this interesting material, such as optoelectronic devices, biomedical systems, and polymer nanocomposites [28–32]. However, the weak non-covalent interaction between CNTs and polymer matrix has limited the development of novel composites [33,34]. The discovery of polydopamine (PDA), inspired by mussel in nature, has provided an ideal candidate for CNT surface modification [35]. The self-polymerized PDA is able to adhere on most solid surfaces and react with thiol or amine functional groups, hence allowing the formation of a variety of copolymers/CNTs composites [35–38]. In this study, we report the preparation of a $CO_2$-responsive composite, which is multiwall carbon nanotubes coated with PDA and polydimethylaminoethyl methacrylate (PDMAEMA).

## 2. Materials and Methods

The following materials were used in this study: multiwall carbon nanotubes, from Nanjing Jicang Nano Technology Co., Ltd. (Nanjin, China); dopamine hydrochloride, 98%, Cupric chloride, 99%, N,N,N′,N″,N″-Pentamethyldiethylenetriamine, 99%, Tris(hydroxymethyl)aminomethane, hydrochloric acid, L-ascorbic acid, dimethylformamide (DMF, analytical reagent), and triethylamine (analytical reagent) from Sinopharm Chemical Reagent Co., Ltd. (Shanghai, China); 2-Bromoisobutyryl bromide (BiBB), 98%, and 2-(Dimethylamino)ethyl methacrylate, 99%, from Aladdin (Shanghai, China). Purified water is supplied by Watsons.

Preparation of MWCNT-PDA: To a 200 mL Tris buffer solution (25 mM), pH = 8.5, 0.4 g dopamine hydrochloride and 0.2 g carbon nanotubes were added with magnetic stirring. The reaction was kept in 25 °C water bath for 24 h then the product was washed with ethanol 3 times and separated by centrifugation. The product was stored in a vacuum oven at 60 °C for 24 h. The MWCNT-PDA were obtained as a black powder.

Preparation of MWCNT-PDA-BiBB: An amount of 20 mg MWCNT-PDA powder was dispersed in 20 mL DMF with 10 min of ultra-sonication. Then, under $N_2$ protection and magnetic stirring, 1 mL triethylamine (7.2 mmol) was added dropwise. After 10 min, a pre-mixed solution of 0.9 mL BiBB (7.2 mmol) in 10 mL DMF, was added by a constant pressure funnel. The reaction was stirred under continuous $N_2$ presence for 24 h at 25 °C. The mixture was repeatedly washed with acetone, ethanol and water, respectively. Then, the product (MWCNT-PDA-BiBB) was placed in a vacuum oven at 60 °C for 24 h.

Preparation of MWCNT-PDA-PDMAEMA: To a three-neck flask and under $N_2$ protection, MWCNT-PDA-BiBB (20 mg), DMAEMA (7.85 g, 0.05 mol), $CuCl_2$ (0.13 g, 0.05 mol), pentamethyldiethylenetriamine (0.173 g, 0.001 mol), and L-ascorbic acid (0.35226 g, 0.002 mol) were added. Then, the temperature was raised to 60 °C and the reaction was kept under $N_2$ protection for 24 h. After the reaction was completed, the product was collected and successively washed with $CH_2Cl_2$, ethanol and water. The MWCNT-PDA-PDMAEMA products were dried in a vacuum oven at 60 °C for 24 h.

Characterization: Room-temperature infrared spectra were recorded on a Shimadzu IRAffinity-1S Fourier transform infrared spectrophotometer. Typically, 20 scans were added at a resolution of 4 $cm^{-1}$. The TEM characterization was performed with FEI-Talos F200x at 200 kV. Agilent Cary 60 UV-Vis Spectrophotometer was used to measure the transmittance of composite and water mixture.

## 3. Results and Discussion

Figure 1 shows the FTIR spectra in the range of 400–4000 $cm^{-1}$ of the as-received MWCNT, MWCNT-PDA, MWCNT-PDA-BiBB, and MWCNT-PDA-PDMAEMA. The MWCNTs have a series of characteristic absorption peaks. The peaks around 3440 $cm^{-1}$ and 3135 $cm^{-1}$ are from OH stretching, which may be related to presence of water or the carboxylic acid group in the tubes. The resolved OH peaks are ascribed to the hydro-

gen bonding and has been observed for different type of MWCNTs [39]. The 1640 cm$^{-1}$ absorption is from the C=O stretching, which indicates the –COOH in the carbon tubes. The carboxylic acid group is most likely from the acid oxidation processing of the carbon nanotubes. The broad absorption around 1100 cm$^{-1}$ is attributed to C–O–C stretching vibration and the 1400 cm$^{-1}$ peak is due to the –CH$_3$ symmetrical angular deformation. There is almost no difference in infrared absorption between MWCNT and MWCNT-PDA. This is probably due to the thin polydopamine film, about 2–3 nm (from TEM results), whereas the penetration depth of IR is in the micrometer range. Therefore, the spectra of MWCNT-PDA is mostly from the bulk carbon nanotube. The immobilization of BiBB was confirmed by the absorption peak at 680 cm$^{-1}$, which is due to the stretching vibration of C–Br in 2-bromo-propionyl groups. After the ATRP reaction, the spectra of the resulting composite show no absorption of carbon bromine bond, suggesting the successful grafting of PDMAEMA on the surface of PDA.

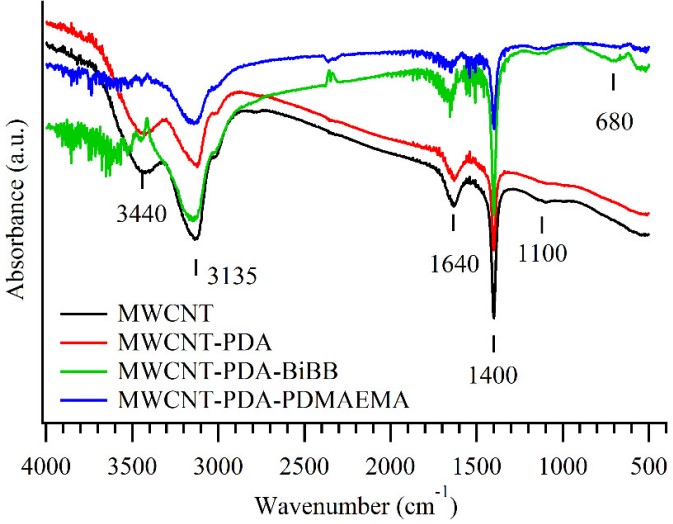

**Figure 1.** FTIR of MWCNTs, MWCNTs-PDA, MWCNTs-PDA-BiBB, and MWCNTs-PDA-PDMAEMA.

The characteristic structure of multiwall carbon nanotubes is observed in Figure 2. The carbon tube has a diameter about 10 nm, and the multiwall extends an additional 5 nm, which provides additional mechanic strength compared to single-walled nanotubes. As shown in Figure 2a, the thin layer, about 2–3 nm on the surface is the PDA, and the walls of nanotubes are intact after the polymer deposition. The thin coating on the surface nanotube thus explains the similarity of IR absorption between MWCNT and MWCNT-PDA. However, the small amount of PDA modification of the MWCNT reacts with BiBB and provides polymerization sites for DMAEMA [40]. The presence of additional material on the surface of MWCNT is confirmed by Figure 2b, showing the thickness of a coating layer about 10 nm. In addition, the MWCNT flakes have low density and float on the surface of water. After coating of PDA, the MWCNT-PDA is still lighter than water; however, the MWCNT-PDA-PDMAEMA composites precipitate and settle to the bottom. To verify the polymerization of truly PDMAEMA, we carry out the CO$_2$-responsive test of this composite.

The multiwall carbon nanotubes tend to aggregate, as indicated in Figure 3. This is due to their strong van der Waals interaction and high aspect ratio. Consequently, the resulting composite is precipitated in water, as shown in Figure 4 inset. However, after bubbling CO$_2$ to the mixture for 5 min, as indicated in Figure 4b, the surface modified MWCNTs turn into well-dispersed in water. This can be explained that the reaction between the PDMAEMA and CO$_2$. The composite becomes hydrophilic as the formation of ammonium bicarbonates. The grafted polymers on the surface of MWCNTs are now positively charged. Therefore, the electrostatic repulsion and steric hindrance leads to the separation of MWCNT aggregates into single tubes [41]. Figure 4 shows the UV-Vis of the

solution before and after the introduction of $CO_2$. Since the MWCNT-PDA-PDMAEMA composite is not dispersed in water, the mixture exhibits hardly any absorption, similar to the absorption of water. However, the bubbling of $CO_2$ leads to the formation of polar compounds and the dispersion of the composite dramatically decreases the transmittance of light; this is due to the dark appearance of the carbon nanotubes.

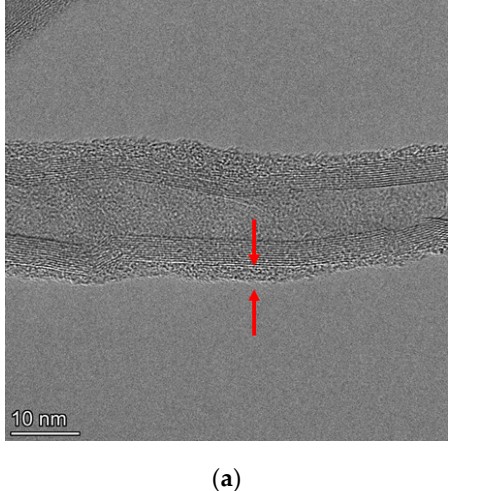 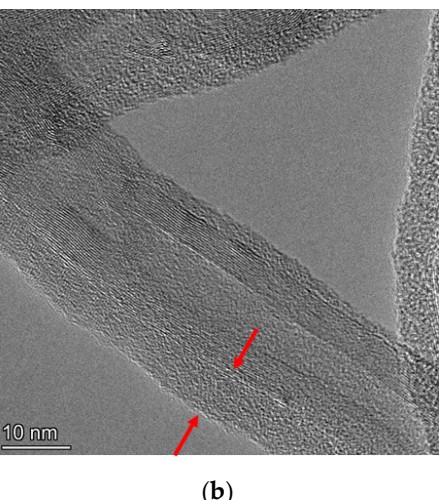

(**a**)                                                    (**b**)

**Figure 2.** TEM of (**a**) MWCNTs-PDA and (**b**) MWCNTs-PDA-PDMAEMA.

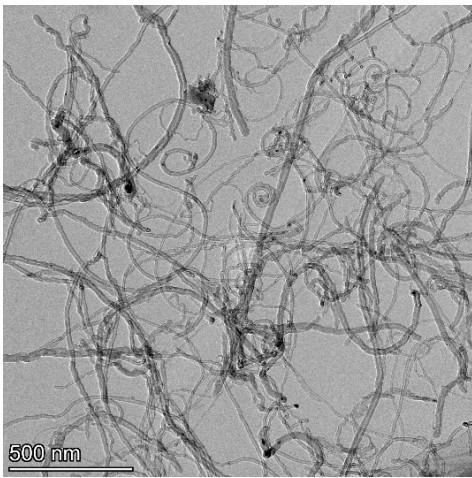

**Figure 3.** TEM of MWCNTs.

To check the reversibility of the composite with $CO_2$, $N_2$ gas was then bubbled into the solution for 10 min. The MWCNT-PDA-PDMAEMA composite precipitated out again and changed back to the original state. The removal of $CO_2$ by $N_2$, reverses the reaction back to the neutral state of PDMAEMA. This results in the increase in the polymer chain interaction and the MWCNTs bundles again, eventually precipitates out of the solution. We propose Scheme 1 for the reversible reaction of the composite with $CO_2$ based on the above observations. This $CO_2/N_2$ purge process was repeated six times and the corresponding UV-Vis spectra were recorded. Using the absorption at 610 nm as an indicator, the cycling results are shown in Figure 5a. This $CO_2$-responsive process renders the composite a potential candidate for many smart material applications. Since the MWCNT-PDA-PDMAEMA composite is responsive to $CO_2$, it is intuitive to think that this material can be used as a $CO_2$ detector. Again, we measure the UV absorption of the mixture against $CO_2$ bubbling time at a fixed flow rate. As shown in Figure 5b, the UV-Vis spectra at 610 nm, the transmittance decrease in the solution has a good correlation as the $CO_2$ gas reacts with the PDMAEMA in 10 min. After that, when the reaction is complete,

the transmittance no longer changes and the curve remains flat. These results show the high sensitivity of composite to the small $CO_2$ stream, which makes it a possible candidate for detecting the presence of $CO_2$ in aqueous solutions.

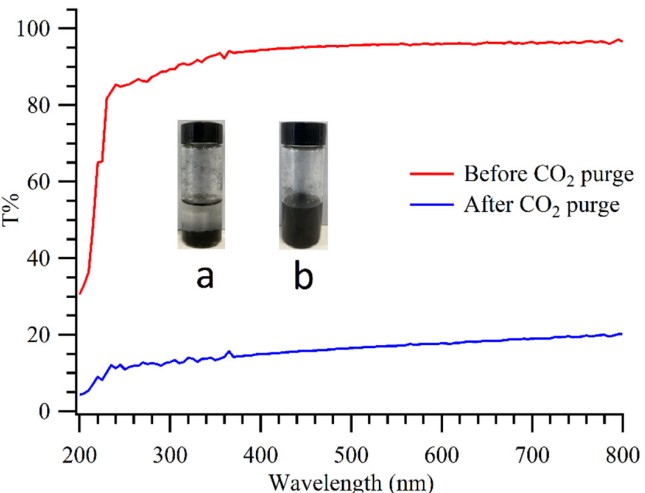

**Figure 4.** UV-Vis spectra of MWCNT-PDA-PDMAEMA in water, inset (**a**) before $CO_2$ purge and (**b**) after $CO_2$ purge.

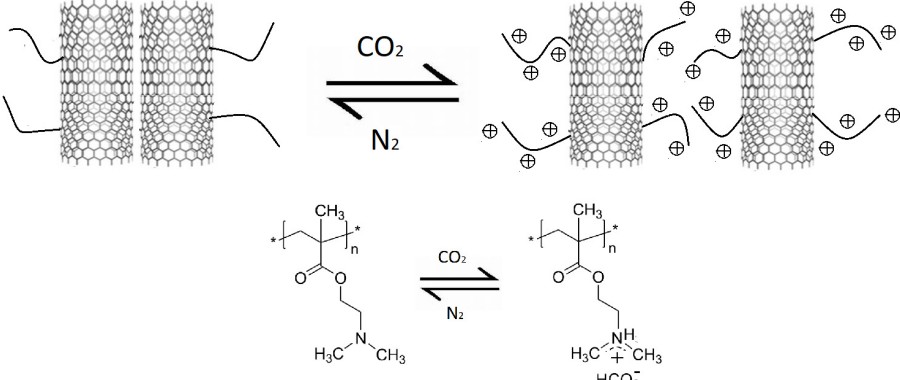

**Scheme 1.** The reaction between composite and $CO_2$ leads to the separation of CNT bundles.

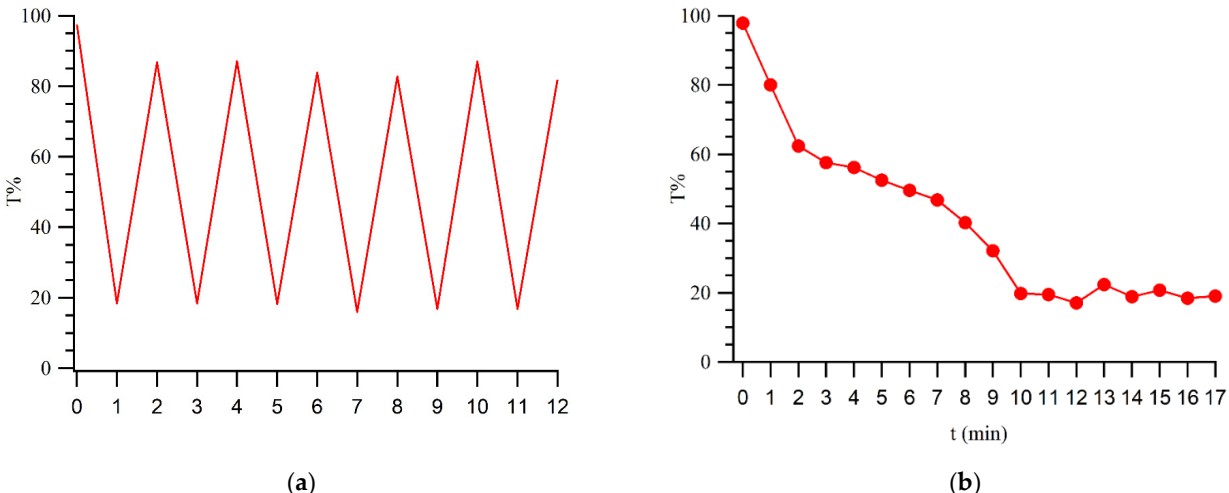

(**a**)　　　　　　　　　　　　　　　(**b**)

**Figure 5.** (**a**) UV-Vis transmittance of composite and water mixture at 610 nm with 6 $CO_2/N_2$ purge cycles; (**b**) UV-Vis transmittance of composite and water mixture at 610 nm against time.

## 4. Conclusions

MWCNTs provide unique mechanical and electrical properties and have a variety of applications for nanotechnology and material science. In this study, a novel $CO_2$-responsive composite was prepared by surface modification of MWCNT. The PDA is coated on the surface of MWCNT due to the presence of highly reactive catechol and amine groups. The large amount of unsaturated active valents of PDA immobilize the BiBB, which initiates the ATRP synthesis of PDMAEMA. The resulting MWCNT-PDA-PDMAEMA composite reversibly reacts with $CO_2$. The forward reaction forms polar ammonium bicarbonate in water, leading to the positive charge accumulation on the surface of MWCNTs. The electrostatic repulsion causes the separation of the carbon nanotube bundle and the composite becomes well dispersed. By bubbling $N_2$ to remove $CO_2$ from the solution, the reverse reaction changes the composite back to the original state and precipitates out. This makes the composite a potential candidate for $CO_2$-responsive smart material application. In addition, the sensitivity of the MWCNT-PDA-PDMAEMA may also be used as a $CO_2$ detector.

**Author Contributions:** Conceptualization, Y.M. and D.H.; methodology, X.C., W.L. and Z.Z.; writing–original draft preparation, Y.M.; writing–review and editing, Y.M. All authors have read and agreed to the published version of the manuscript.

**Funding:** This research was funded by Innovative Research Team of Taizhou Polytechnic College (No. TZYTD-16-4) and the Excellent Young Teachers of Jiangsu Blue Project, from Taizhou Science and Technology Support Program (Project TN202012).

**Institutional Review Board Statement:** Not applicable.

**Informed Consent Statement:** Not applicable.

**Data Availability Statement:** The data presented in this study are available in the main text of the article.

**Conflicts of Interest:** The authors declare no conflict of interest.

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
