# Peer review of "Preparation of a Novel CO2-Responsive Polymer/Multiwall Carbon Nanotube Composite"

_processes, doi:10.3390/pr9091638_

Round 1
Reviewer 1 Report
Correction
1- Correct sentence "Then the temperature was then raised 60 °C, and under the N2 protection, the reaction was kept for 24 h"
2-Correct sentence "After the reaction was completed. the product was successively washed with CH2Cl2, ethanol and water."
3- Write full abbreviation of DMF and BiBB
Questions
Q 1. There is no data to compare separately the effect of CO2 on MWCNTS and composite MWCNTs-PDA-PDMAEMA. Show the effect of CO2 bubbling on MWCNTs without making a composite.
Q 2. How much is settling time prior to UV-Vis measurement for MWCNT-PDA-PDMAEMA before and after CO2 purge?
Q 3. How is the effect of sonication on the dispersion of composite compare to CO2 bubbling for the same settling time?
Q 4. How does reaction become reversible with the addition of N2?
what is the basis for Scheme 1?
Author Response
Response to Reviewer 1 Comments and Questions
Comment 1: Correct sentence "Then the temperature was then raised 60 °C, and under the N2 protection, the reaction was kept for 24 h" 

Response 1: The sentence has been changed to "Then the temperature was raised to 60 °C and the reaction was kept under N2 protection for 24 hours."
Comment 2: Correct sentence "After the reaction was completed. the product was successively washed with CH2Cl2, ethanol and water."
Response 2: The sentence has been changed to "After the reaction was completed, the product was collected and successively washed with CH2Cl2, ethanol and water."
Comment 3: Write full abbreviation of DMF and BiBB
Response 3: Dimethylformamide (DMF) was added to the Material section. BiBB was abbreviated in the Abstract section and now also added in the Material section.
Q1. There is no data to compare separately the effect of CO2 on MWCNTS and composite MWCNTs-PDA-PDMAEMA. Show the effect of CO2 bubbling on MWCNTs without making a composite.
Answer 1: The MWCNT flakes float on the surface of water and there is no reaction after CO2 bubbling. And the description has been added in the Results section.
Q2. How much is settling time prior to UV-Vis measurement for MWCNT-PDA-PDMAEMA before and after CO2 purge?
Answer 2: The UV-Vis measurement was conducted right after the purge and there was no settling time.
Q3. How is the effect of sonication on the dispersion of composite compare to CO2 bubbling for the same settling time?
Answer 3: After sonication, the composite just started to precipitate, however, after the CO2 bubbling, the composite dispersed well and formed a stable system.
Q 4. How does reaction become reversible with the addition of N2? What is the basis for Scheme 1?
While bubbling N2 or other inert gas (Ar) into the solution, the deprotonation of bicarbonate salts of tertiary amines takes place rapidly. With the formation of CO2, PDMAEMA changes back to the original neutral state, thus the reaction is reversed. This has been a standard test for amine based materials that are CO2 responsive.

Reviewer 2 Report
This paper is just copy of many papers reported by other researches although authors use the multiwall carbon nanotubes (WCNT). Reviewer does not feel scientific meaning. Moreover, reviewer cannot be confirmed that the polymerization of DMAEMA proceeded on the basis of spectral changes by Figure 1 (FTIR). Authors mentioned in the text that thickness of polymer (PDMAEMA) layer on the MWCNTs-PDA expanded from 2~3 nm (PDA layer) to 10 nm (see text) on polymerization by TEM measurement. If so, almost MWCNTs-PDA are covered with PDMAEMA polymer brushes (see TEM). However, the carbonyl peak (1655 cm-1) in the FTIR of Fig.2 is very small. The result of TEM is not agreement with the FTIR result. Reviewer feels that detailed characterization of surfaces at the reaction stages starting from MWCNT are required. Unfortunately, this paper is not acceptable at present style for publication in Processes. BiBB is what.
Author Response
Response to Reviewer 2 Comments
Comment 1: This paper is just copy of many papers reported by other researches although authors use the multiwall carbon nanotubes (WCNT). Reviewer does not feel scientific meaning.
Response 1: Many scientific investigations have similar experimental designs to tackle the same scientific challenge. The authors have not found the same copy of our current study according to our reference search. The reviewer also mentioned the use of multiwall carbon nanotubes. Since it has better mechanical strength and other advantages than single wall carbon nanotubes and some other nano materials, it is interesting to see the difference. We have prepared a novel composite and believe there is scientific meaning for this report.
Comment 2: Moreover, reviewer cannot be confirmed that the polymerization of DMAEMA proceeded on the basis of spectral changes by Figure 1 (FTIR). Authors mentioned in the text that thickness of polymer (PDMAEMA) layer on the MWCNTs-PDA expanded from 2~3 nm (PDA layer) to 10 nm (see text) on polymerization by TEM measurement. If so, almost MWCNTs-PDA are covered with PDMAEMA polymer brushes (see TEM). However, the carbonyl peak (1655 cm-1) in the FTIR of Fig.2 is very small. The result of TEM is not agreement with the FTIR result. Reviewer feels that detailed characterization of surfaces at the reaction stages starting from MWCNT are required.
Response 2: We agree with the reviewer that FTIR is not the best tool to characterize the nano-scale surface film. As we have mentioned in the report, the electromagnetic radiation in the IR range can penetrate deep. The information is mostly from the bulk carbon nanotube, as we can see the similarity from all the spectra. The carbonyl absorption is mostly from the COOH group, which is formed during the acid oxidation. The confirmation of the initiator and PDMAEMA is based on the observation of C-Br (680 cm-1). Another circumstantial evidence is that the MWCNT and MWCNT-PDA are both lighter than water and they float on the surface. After the synthesis of MWCNT-PDA-PDMAEMA, the composite precipitates. We have added this description in the Results section. In addition, the CO2 responsive results are direct evidence of the polymerization of PDMAEMA on the surface. We have also conducted TEM EDX measurements on these materials, however, the C, O, and N content cannot differentiate the difference between PDA and PDMAEMA, and therefore they are not included in the report.
Comment 3: BiBB is what.
Response 3: BiBB is 2-bromoisobutyryl bromide and it was abbreviated in the Abstract section. Now it is also added to the Material section.

Round 2
Reviewer 1 Report
The manuscript is fine now.
Author Response
The authors thank the reviewer for the professional and kind comments.
Reviewer 2 Report
As to Authors Response for Comment 1: Authors mention that this report has a scientific meaning in the point of revealing the difference between multi wall CNT (MWCNT) composite and single wall CNT (SWCNT) one. If so, what is significant differences between MWCNT-based composite and SWCNT-based one which authors discovered in these experiments. Authors do not mention this point in the text, and should describe this point. It is generally known that there is the difference in mechanical strength between MWCNT and SWCNT.
As to Response for Comment 2: EPS measurements will support this events. See a following paper: Q. Wan et al., J. Polym. Sci., Part A Polym. Chem. 2015, 53, 1872-1879.
As to Response for Comment 3: OK.
Author Response
As to Authors Response for Comment 1: Authors mention that this report has a scientific meaning in the point of revealing the difference between multi wall CNT (MWCNT) composite and single wall CNT (SWCNT) one. If so, what is significant differences between MWCNT-based composite and SWCNT-based one which authors discovered in these experiments. Authors do not mention this point in the text, and should describe this point. It is generally known that there is the difference in mechanical strength between MWCNT and SWCNT.
Response: The report is not about the difference between SWCNT and MWCNT. The mention of different from SWCNT was due to the reviewer thought our report was a copy except using MWCNT, which we could not agree. The scope of our study is to prepare a composite with MWCNT and CO2 responsive polymer. And comparison between SWCNT and MWCNT could be the next step of our investigation. As for the clear difference, when applications with higher mechanical strength is needed, the MWCNT based composite will reasonably be a better choice.
As to Response for Comment 2: EPS measurements will support this events. See a following paper: Q. Wan et al., J. Polym. Sci., Part A Polym. Chem. 2015, 53, 1872-1879.
Response: After reading the reference, the authors think the reviewer referred to XPS, and we agree that XPS measurement provides the difference state of element on the surface. In our TEM-EDX results (also using X-ray), we also see the difference of element contents. However, it is very difficult to differentiate dopamine (C8H11NO2) and dmaema (C7H13NO2). Therefore we did not include the EDX results. In addition, we have no resource to conduct XPS experiments.